# A Comparative Study of the Fatigue of the Lower Extremities According to the Type of Shoes Worn When Firing a 10 m Air Pistol

**DOI:** 10.3390/ijerph20021363

**Published:** 2023-01-12

**Authors:** Yuxi Zhang, Ze Zhang, Sukwon Kim, Youngsuk Kim

**Affiliations:** Department of Physical Education, Jeonbuk National University, Jeonju 54896, Jeollabuk-do, Republic of Korea

**Keywords:** shoes, air pistol, biomechanics, surface EMG

## Abstract

The purpose of this study was to compare the differences in shooting performance, physical stability, and lower extremity muscle fatigue in high-level shooters wearing shooting shoes and sports shoes, and to determine the causes. Eight high-level 10 m air pistol shooters wore shooting shoes and sports shoes in a simulated shooting qualification experiment (60 rounds in 75 min) and we recorded shooting scores, pressure centers (COP), and median frequency of muscle fatigue index (MDF). All the data are expressed as mean ± standard deviation and are compared using a paired *t*-test. Athletes in shooting shoes scored higher than those in sports shoes (*p* < 0.05); COP moved less in the front-to-back and left-to-right directions for athletes wearing shooting shoes rather than sports shoes; and fewer muscles experienced fatigue, with fatigue also occurring later, for athletes wearing shooting shoes rather than sports shoes. Shooting shoes may reduce the sway of athletes’ center of pressure in the anterior–posterior and left–right directions, enhance postural stability, and result in higher shooting scores. In addition, they may make shooters feel more comfortable and relaxed during long training sessions and competitions.

## 1. Introduction

Competitive shooting is a sport that uses guns and bullets to shoot at preset targets. The number of rings (or targets) hit is used to calculate the score. The sport is characterized by the fact that each participant has their own target, relying on their own technical level and mental control ability to complete the competition and seek to win it. Thus, it can be seen that the evaluation of good shooting is determined by shooting performance (the number of rings or targets hit).

Studies have shown that shooting sports are static sports in terms of muscle exertion, and athletes have to maintain an action for a certain time [1]. In particular, the pistol event—owing to the small size of the pistol, its light weight, and its grip position at the end of the limb, that is, the position of the hand—has fewer support points and the worst stability, making the gun very difficult to stabilize from a point of immobility. Wan believes that the basic action of pistol shooting is mainly based on the aiming and firing of the gun [2]. These actions are interconnected. Aiming is the basis of all the movements. Without a stable gun, it is impossible to aim accurately and fire correctly and freely. The target size is very small (11.5 mm ± 0.1 mm in diameter for a 10-point maximum score) over a shooting distance of 10 m [3]. Therefore, angular deviations as small as 0.066° may result in significant score losses in high-level pistol shooting. Although many factors influence air pistol shooting performance, the ability to stabilize the gun appears to play the most important role [4].

Studies have shown that the ability to stabilize the shot and maintain excellent shooting performance is controlled by the movement of the body’s center of pressure (COP) [5]. There is a significant correlation between shooter performance and COP exercise [6]. Therefore, in order to stabilize the gun to a greater extent, the shooter needs maximum control over body movements. In other words, better static balancing leads to a higher level of performance. When a person is in an upright standing position, the integrity of the standing posture is maintained through sustained corrective contractions of the lower limb and trunk muscles in multiple directions [7]. Therefore, in shooting events, standing for a long time can lead to prolonged muscle contractions, which create fatigue and affect balance and stability.

At present, the study of fatigue mainly utilizes surface EMG detection techniques, and a relatively well-developed theoretical system has basically been formed. It has been found that the EMG power spectrum shifts to the left when muscle fatigue occurs, i.e., the low-frequency component increases and the high-frequency component decreases [8]. Shooting is a type of static sport, with shooting action, balance, stability, and consistency derived from muscle contractions. Shooting accuracy is thus the external performance of fine muscle movements. In particular, shooting requires precision, accuracy, and stability, so the proprioception of muscles, and the stability and consistency of force, are very important.

Studies have shown that shoes can affect a person’s balance and stability, resulting in changes in COP. The hardness of the sole is an important factor affecting the function of the shoe, and shooting shoes are harder than ordinary sports shoes. Prior research shows that the different hardness of shoe soles has a great influence on the ankle joint and the first metatarsophalangeal joint. Increasing the mid-sole hardness of shoes can reduce the maximum ankle dorsiflexion angle [9,10], increase stability and reduce fatigue [11]. When fatigue is produced, the maximum distance that COP moves in the x and y axes increases, and the average or maximum velocity in the x and y axes increases [12].

At present, the research status of shooting focuses on the psychological level, injury level, and other aspects. In the field of biomechanics, the study of shooting sports mainly focuses on laser testing systems and surface myoelectric signals of gun arms. The role of shoes in influencing balance and stability is enormous, affecting the activity of the lower extremity muscles and impacting on biomechanical indicators such as COP and surface electromyogram signals. At present, there is no literature showing the biomechanical effects of shoe changes on shooting sports. Therefore, according to shooting rules, this study investigates the differences in electromyography, balance, stability, and shooting performance of lower limb muscle surfaces between shooting shoes and sports shoes. This study provides a theoretical basis for coaches and athletes to influence shooting sports through the use of shooting shoes.

## 2. Methods

### 2.1. Subjects

Eight experimental subjects were selected for this study to analyze the effect of wearing different shoes on shooting performance, body center of pressure and surface EMG indices. For basic information on all the shooters, please refer to Table 1. The inclusion criteria for the study were: (1) holding a gun in the right hand; (2) no history of injury to the trunk, spine, and limbs, no restricted joint range of motion, and no muscle weakness; and (3) experience in participating in competitions. Subjects were informed of the purpose of the experiment and the procedures, it was ensured that they understood the aim of the experiment and that they were participating voluntarily, and they signed a consent form. To remove order effect, randomization was performed.

### 2.2. Experimental Procedure and Apparatus

Participants were asked to stand 10 m away from a paper target for an experiment that simulated a qualification match, and to commission the equipment before starting the experiment with 5 empty shots, which were not counted. After starting the experiment, participants were given 60 shots within 75 min, during which they could follow their personal shooting habits, such as taking breaks and drinking water. Participants used their own firearms, shooting shoes, sports shoes, etc. to conduct the experiment to ensure that they could perform at their best level.

For test data collection, shot scores (with decimals) were measured using a SCATT optical sensor (SCATT MX-W2, part number: 101708), a training device consisting of an optical transmitter and receiver unit (weighing 0.45 kg). The SCATT training device was connected to a laptop computer on which the shot scores were displayed for the participants. Each shot (60 shots per participant for a total of 960 test shots) was recorded with a shot score at a sampling rate of 120 Hz. During the test, only the shot scores and hit locations were displayed for the participants, similar to a genuine competition situation. Sighting ballistic data were displayed to the participants afterwards to avoid influencing their shooting technique during the test.

The kinematic data of each participant were captured using 13 infrared cameras (OptiTrack, Natural Point, Inc., Corvallis, OR, USA) at a sampling rate of 120 Hz. In the experiment, the matched marker was a 14 mm reflective marker and each subject had 28 reflective skin markers, including 18 bone markers, 6 calibration markers and 4 markers for thigh and calf sections [13,14].

The OR6-6-2000 force platform (AMTI, Inc., Newton, MD, USA) was used to collect ground reaction force data at 1200 Hz with a maximum delay time of 6 ms.

The EMG acquisition system (Trigno Avanti Sensor, Delsys, Natick, MA, USA) was selected as the EMG data acquisition device. For EMG signal acquisition, we used the Trigno Avanti Sensor (Delsys, Natick, MA, USA; 3.7 cm × 2.7 cm). All EMG sensors had a common mode rejection ratio of 80 dB and were synchronized with kinematic and kinetic data by recording Motive software (OptiTrack, Natural Point, Inc., Corvallis, OR, USA) with an EMG sampling frequency of 1200 Hz.

Before attaching the electrodes, with the consent of the subjects, the skin surface was shaved and washed with alcohol, and after the skin was dry, the EMG electrodes were attached. The electrodes were also fixed with motion tape to reduce interference signals [15].

In the experiment, we mainly tested 8 muscles of the lower limbs (Table 2).

Subjects were asked to fire 60 rounds of non-live ammunition in 75 min, during which period they followed their own shooting habits. Subjects took each shot after hearing the command: “Ready, 5-4-3-2-1, go”. The data for the 1st, 30th, and 60th shots were collected for subsequent analysis.

### 2.3. Data Processing and Analysis

A SCATT optical sensor (SCATT MX-W2, part number: 101708) was used to measure the shooting scores (with decimals), and the 1st, 30th, and 60th shot scores of each participant wearing normal sports shoes, and the 1st, 30th, and 60th shot scores of each participant wearing shooting shoes were selected.

In this study, an infrared motion capture system (acquisition frequency of 360 Hz), an AMTI 3D force platform (acquisition frequency of 1200 Hz), a surface EMG system (acquisition frequency of 1200 Hz), and a laser targeting system (SCATT MX-W2) were used to acquire kinematic, kinetic, surface EMG, and shooting performance data for analysis.

We used the motion capture system software to pre-process the data (muzzle and trigger position marker point name, surface EMG sensor name, etc.), and the two-point distance formula in the 3D coordinate system to calculate the maximum displacement value of the muzzle and trigger position marker, so as to determine the specific number of frames at the moment of firing. Before firing, stability is directly related to shooting performance. Therefore, the 3 s before firing is crucial. The data were taken 3 s before firing, so the specific number of frames at the moment of firing was subtracted from 360 frames to determine the data analysis interval as follows:(1)$$\sqrt{\left[{\left(x1-x2\right)}^{2}+{\left(y1-y2\right)}^{2}+{\left(z1-z2\right)}^{2}\right]}$$

The distance between two points in a three-dimensional coordinate system.

The surface EMG data were first filtered using a Butterworth 4th order filter, and the band pass filter was selected with a window value of 10 Hz–400 Hz. The surface EMG data for 3 s before firing were processed using median frequency (MDF), which is an important index for judging fatigue value.

After the data were collected using the motion capture system software, the raw data collected in Motive were exported in C3D file format, and the C3D files were imported into Visual 3D for modeling to complete the body COP analysis. The mean of the analyzed COP was plotted into a line graph using Origin 2019b 64 Bit graphing software.

The data obtained were calculated and statistical analysis of the relevant data was performed using the GraphPad Prism 9 statistical software package. The statistical descriptions were calculated using the mean ± standard deviation and the Shapiro–Wilks test was performed as a normality test. When the normality of the data was satisfied, a t-test was performed to verify the difference between the two treatments, with *p* < 0.05 as significant and *p* < 0.01 as highly significant.

## 3. Results

### 3.1. Shooting Performance

As can be seen from Table 3, there was a significant difference in the shooting scores of S3 and S7 when they were shooting in different shoes, and no difference in the shooting scores of the remaining six participants whether they were wearing shooting shoes or sports shoes. There was a significant difference in the mean shooting scores of the eight subjects depending on whether they were wearing shooting shoes or sports shoes, with higher scores for those wearing shooting shoes than for those wearing sports shoes.

### 3.2. Center of Pressure (COP)

The COP *x*-axis represents the change in distance traveled by the body’s center of pressure in both the left and right directions. As can be seen in Figure 1, the range of movement is smaller when wearing shooting shoes than when wearing sports shoes.

The COP y-axis represents the change in distance traveled by the body’s center of pressure in the anterior–posterior direction. As can be seen from the graph, shooting shoes move less than sports shoes and show less significant swaying.

As seen in Table 4, there was no significant difference in the trajectory of COP movement in the anterior–posterior (*x*-axis) and left–right (*y*-axis) directions whether participants were wearing shooting shoes or sports shoes (*Px* = 0.362, *Py* = 0.911). However, the mean and standard deviation of the trajectory in the x and y axes for those wearing shooting shoes were slightly smaller than for those wearing sports shoes.

### 3.3. Muscle Fatigue

The results showed that there was a significant difference in the MDF of the left anterior tibialis muscle depending on the type of shoe being worn, while there was no significant difference in the MDF of the right anterior tibialis muscle. The MDF of the left tibialis anterior muscle showed a decreasing trend with time for shooters wearing sports shoes, while the MDF of the left tibialis anterior muscle did not change significantly with time for shooters wearing shooting shoes (Figure 2).

The results showed no statistically significant difference in either side of the gastrocnemius muscle whether participants were wearing shooting shoes or sports shoes (*p* > 0.05) (Figure 3).

The results showed that there was a significant difference in the MDF of the rectus femoris muscle on both sides of the shooters when wearing shooting shoes and sports shoes. The MDF of both sides of the rectus femoris muscle when participants were wearing different shoes showed a trend of decreasing Hertzian numbers with time (Figure 4).

Figure 5 shows that there was a significant difference in the MDF of the semitendinosus muscle on both sides of the shooters when wearing shooting shoes and sports shoes. Among participants wearing sports shoes, the MDF Hertz number for both sides of the semitendinosus muscle showed a decreasing trend with increasing time.

As can be seen from Table 5, a decreasing trend in the MDF Hertz number for the left anterior tibialis and both rectus femoris muscles was observed between the 30th and 60th shots when shooting shoes were worn.

When participants wore sports shoes, between D1 and D60 there was a decreasing trend in the MDF Hertz number of the left anterior tibial muscle, the left rectus femoris and the right semitendinosus; and between D30 and D60 there was a decreasing trend in the MDF Hertz number of the right rectus femoris and the left semitendinosus (Table 5).

## 4. Discussion

### 4.1. Shooting Performance

A significant difference was noted in shooting performance between participants wearing shooting shoes and those wearing sports shoes. Shooting sports require high standards of consistency, regularity, and stability of technical movements. In terms of technical action, if each shot can maintain the same gun action, the firing action is regular, and 10 rings can be hit in a row, this is a guarantee of achieving high performance [12]. In actual sports training, the purpose is to consolidate correct movements, strengthen the muscular sensation of the correct movement, minimize the shaking of the gun before shooting, and improve the ability to hit 10 rings in a row. As can be seen from Table 3, although two athletes’ scores differed depending on which shoes they were wearing, among the remaining athletes there was no difference in shooting scores between the two types of shoe, which may be because excellent athletes have better specialized strength, and more accurate and consistent movements, so they can rely on their excellent shooting techniques and reduce the influence of shoes on shooting rings. In terms of the overall mean and standard deviation of shooting scores, the scores for athletes wearing shooting shoes were higher than for those wearing sports shoes, and in shooting competitions, even a 0.1-point difference in scores is a key factor in determining victory or defeat, so the role of shooting shoes can thus be considered significant.

### 4.2. Center of Pressure (COP)

COP is an important indicator to illustrate balance and stability. Balance and stability refer to the ability to maintain body posture, in particular the ability to control the body’s center of gravity and center of pressure on a small support surface. The balance type required by shooting athletes is static balance, and high-level athletes have reached a certain level after lengthy professional training. The COP x-axis column data and *P_x_* values in Table 4 show that the sway of the center of pressure in the left and right directions of the shooting athletes in the firing process is small, and paired *t*-test analysis of the COP left and right directional sway data of eight shooting athletes wearing shooting shoes and sports shoes shows no significant difference. This indicates that the degree of center-of-pressure sway in the left–right direction was not affected by the shoe factor. Regarding the action posture analysis, in the left–right direction, athletes stand with their feet apart, placed slightly wider apart than their shoulders, and the support surface for balance is relatively large, which is conducive to the control of body balance. As can be seen from Figure 1, the range of left and right movement is smaller when wearing shooting shoes than when wearing sports shoes. This may be due to the stiffer middle sole and material of shooting shoes, which reduces the movement angle of the ankle joint and thus improves stability. Combined with the analysis of shooting posture, the shooters’ feet are opened in the left and right directions, slightly wider apart than the shoulders, and the balanced supporting surface is relatively large, which is conducive to the control of body balance. This may be the reason why the shaking range of different shoes on COP in the left and right directions is small and there is no significant difference between the shoe types.

The COP *y*-axis data and *P_y_* values showed that the center of pressure swayed less in the anterior–posterior direction for shooters wearing different shoes during the firing process, and paired *t*-test analysis of the sway data in the anterior–posterior direction for shooters wearing different shoes showed no significant difference (Table 4). This indicates that the center of pressure sway in the anterior–posterior direction was not influenced by the shoe factor. As can be seen from Figure 1, COP for those wearing shooting shoes had a smaller shaking range in the forward and backward direction than COP for those wearing sports shoes. It can be seen from Table 4 that COP slosh on the x-axis is less than that on the y-axis, which is consistent with the research of Liu Min et al. [16]. This may be because shooting shoes are high in the back and low in the front, and there are straps in the middle of the shoes to reduce the shaking of the foot in the shoe, so shooting shoes provide more stable support in the front and back direction of the center of pressure than sports shoes. According to the postural analysis, a shooter’s aiming process is from down to up and then to the center of the target. Therefore, from 3 s before the launch to the launch, a high-level shooter has adjusted the muzzle of their gun to a very precise position and completed the relative static state. Therefore, the COP wobbles less in the forward and backward direction than in the left and right direction.

Although no statistical difference was noted in the center of pressure in the anterior–posterior and left–right directions with the change of shoes, the range of sway was always smaller in shooting shoes than in sports shoes in terms of movement values, suggesting that shooting shoes also have a better effect on the balance and stability of shooters.

### 4.3. Muscle Fatigue

A prerequisite for shooting athletes wishing to obtain excellent results is a large amount of training. During shooting competitions, athletes also need to stand for about 75 min. An increase in training intensity and time will produce muscle fatigue. Therefore, the fatigue level of lower limb muscles during shooting can be used to indicate whether shooting shoes are helpful for shooting athletes who stand for long periods of time. Several muscle fatigue features based on surface EMG signals have been widely used, and different feature parameters have different effects on the evaluation of fatigue. In this study, the median frequency (MDF) parameter was selected to evaluate the effect of shooting shoes and sports shoes on lower limb fatigue. If the MDF index shows a decreasing trend with time, where the low frequency component increases and the high frequency component decreases, it implies fatigue [8].

From an anatomical point of view, the rectus femoris is an important muscle that allows calf extension, thigh extension and flexion, knee extension, and hip flexion, and that maintains the standing posture of the body and knee stability. Maintaining postural integrity is achieved by corrective contraction of the muscles relative to the lower extremity, moving body weight in multiple directions. Therefore, there is variability in the MDF index of the rectus femoris muscle regardless of whether shooting shoes or sports shoes are worn, and there is a decreasing trend over time. As seen in Table 5, the MDF Hertzian number of the anterior tibialis muscle on both sides decreased between the 30th and 60th shots when wearing shooting shoes, representing the development of fatigue.

The semitendinosus muscle can flex the knee joint, extend the hip joint, and rotate the lower leg inward when flexing the knee. The shooting action requires keeping the knee joint hyperextended, so there is a difference in the MDF index of the semitendinosus muscle in different shoes. From Figure 5, it can be seen that when athletes wore shooting shoes, the semitendinosus muscle did not show a decreasing trend in MDF during the shooting process, and fatigue was not produced; however, when wearing sports shoes, both sides of the semitendinosus muscle showed fatigue, and MDF showed a decreasing trend. As can be seen from Table 5, the decrease in MDF of the left semitendinosus occurred between the 30th and 60th shots, while the decrease in MDF of the right semitendinosus occurred between the 1st and 60th shots.

The anterior tibialis muscle makes the foot dorsiflex and turns the foot inward; when the foot bone is fixed together with other muscles, contracting it can make the lower leg tilt forward. From the analysis of action posture, shooters in the shooting process with feet apart, lower legs tilted slightly forward, the body slightly turned to the right and collapsed, and the upper body slightly tilted to the left rear, the whole body’s center of gravity falls in the center of the support surface or slightly to the left front [17,18], which may be the reason for the significant difference in the left anterior tibialis MDF between athletes wearing the different shoe types. From Table 5, it can be seen that when wearing shooting shoes, the decrease in MDF of the left anterior tibial muscle appeared between the 30th and 60th shots; while, when wearing sports shoes, the Hertzian number of MDF of the left anterior tibial muscle gradually decreased between the 1st and 60th shots. The design of the heel of shooting shoes reduces the muscle force of the anterior tibial muscle during the forward tilt of the lower leg, which may explain the later appearance of the decreasing trend of MDF of the left anterior tibial muscle when wearing shooting shoes.

## 5. Conclusions

The biomechanical variability of good shooters while wearing shooting shoes and sports shoes was investigated, mainly divided into kinematics, kinetics, laser targeting equipment, and some related lower limb muscle aspects for comparison. This study combined the use of the Motive motion capture system, the C3D data analysis system, the SCATT laser targeting system and the surface myoelectric system to capture motion data more accurately and easily, and to make the experimental results more accurate. The height, weight, and age of the subjects in this study were not significantly different, which excluded the influence of external variables and made the data more valid.

(1) The shooting action is not merely a muscular sensation, but also has significant requirements relating to the speed of the falling gun, its angle in relation to the target, the time of aiming, the force used to press the trigger, the holding of the excitation moment, and psychology. The change of shoes does not have an effect on some high-level shooters, which may be due to the sporting skills already developed. Overall, it is important to note that shooting performance is better when wearing shooting shoes than when wearing sports shoes.

(2) In the process of shooting, the pressure center of the body swayed less in the front and back direction and left to right direction, and the amplitude of the pressure center sway in the front and back direction was greater than that in the left and right direction, which was consistent with Liu Min’s research results [16]; when the shooters wore sports shoes, the center of pressure of the body swayed more in the front-to-back and left–right directions than in the shooting shoes.

(3) During shooting, depending on the shoe, the main muscles affected are the rectus femoris, semitendinosus, and left tibialis anterior; according to the results, shooting shoes can reduce the number of muscles that appear fatigued and delay the onset of fatigue. Therefore, shooting shoes may enable shooters to feel more comfortable and relaxed during lengthy training and competitions.

(4) Regarding future research direction, this study only compares and analyzes the difference between two kinds of shoes on various shooting indicators, but does not study their correlation. In the future, the correlation could be studied to better understand the effect of shooting shoes on shooting sports, so as to provide more effective guidance to shooters and coaches.

(5) In terms of limitations of the results, the subjects in this study were all shooters at a high level, and the sample size was small, which may not represent the general phenomenon of the whole shooting group. Psychological factors are also among the factors that have a great influence on shooting sports. Therefore, the influence of athletes’ psychological factors on the results cannot be ruled out. In terms of experimental design, in order to reduce the influence of fatigue on the results, the shooters were allowed to take a three-day rest after the shooting shoes experiment. On the contrary, they were more familiar with the experimental environment and content when wearing sports shoes than when wearing shooting shoes, which may have influenced the results.

(6) Regarding practical application of the results, by analyzing the differences in various biomechanical characteristics of shooters wearing special shooting shoes and general sports shoes, this study clarifies the good influence of shooting shoes on shooters, which can provide a reference point and theoretical guidance for coaches and shooters in daily training and competition. At the same time, it provides a clear understanding of the role of shooting shoes, so as to provide ideas for future generations to upgrade the performance of shooting shoes.

## Figures and Tables

**Figure 1 ijerph-20-01363-f001:**
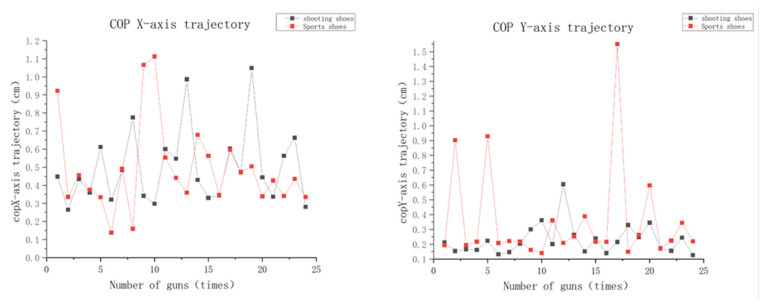
Variation of center of pressure (COP)(x) and center of pressure (COP)(y) travel distance for wearing of shooting shoes and sports shoes.

**Figure 2 ijerph-20-01363-f002:**
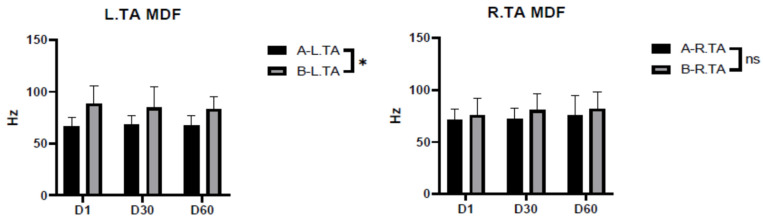
Median frequency (MDF) of the tibialis anterior muscle while wearing shooting shoes and while wearing sports shoes. Note: ns indicates *p* > 0.05, which is no significant difference; * indicates *p* < 0.05, which is significant; A indicates shooting shoes; B indicates sports shoes; L indicates left; R indicates right; TA indicates tibialis anterior muscle; and D1/D30/D60 indicate times.

**Figure 3 ijerph-20-01363-f003:**
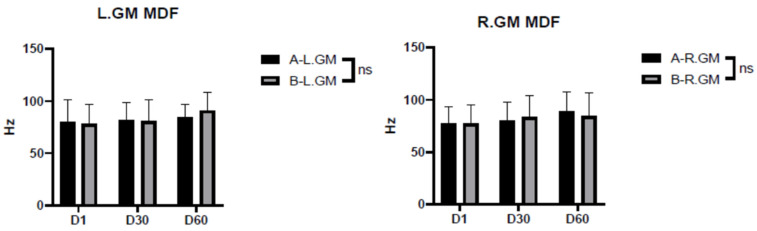
Median frequency (MDF) of the gastrocnemius muscle while wearing shooting shoes and while wearing sports shoes. Note: ns indicates *p* > 0.05, which is no significant difference; A indicates shooting shoes; B indicates sports shoes; L indicates left; R indicates right; GM indicates castrocnemius muscle; and D1/D30/D60 indicate times.

**Figure 4 ijerph-20-01363-f004:**
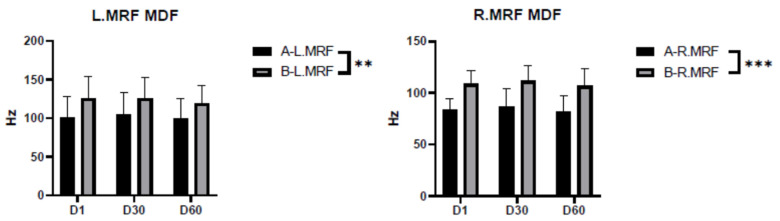
Median frequency (MDF) of the musculus rectus femoris while wearing shooting shoes and while wearing sports shoes. Note: ** indicates *p* < 0.01, which is highly significant; *** indicates *p* < 0.001; A indicates shooting shoes; B indicates sports shoes; L indicates left; R indicates right; MRF indicates musculus rectus femoris; and D1/D30/D60 indicate times.

**Figure 5 ijerph-20-01363-f005:**
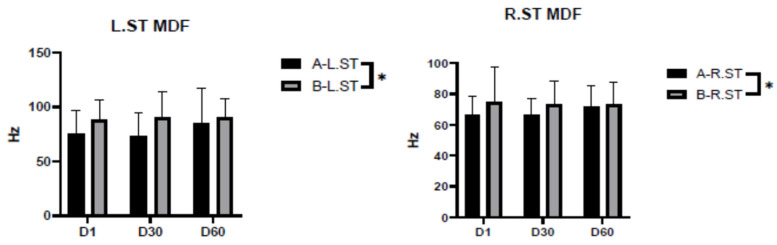
Median frequency (MDF) of the semitendinosus while wearing shooting shoes and while wearing sports shoes. Note: * indicates *p* < 0.05, which is significant; A indicates shooting shoes; B indicates sports shoes; L indicates left; R indicates right; ST indicates semitendinosus; and D1/D30/D60 indicate times.

**Table 1 ijerph-20-01363-t001:** Information on subjects.

N	AGE	Height (cm)	Weight (kg)	Training Years
8	22 ± 2.06	169.13 ± 6.25	67 ± 10.01	7.13 ± 1.83

**Table 2 ijerph-20-01363-t002:** Muscle selection name and position.

Number	Muscle Name	Body Parts	Sensor Pastes Parts
1	Left anterior tibialis muscle (L.TA)	Small head of the fibula to the upper 1/3 of the medial ankle measurement	Tibial trochanter to 20% of the medial ankle
2	Right tibialis anterior muscle (R.TA)		
3	Left rectus femoris (L.MRF)	Anterior superior iliac spine to the superior border of the patella 1/2	50% of the anterior superior iliac spine to the superior border of the patella
4	Right rectus femoris (R.MRF)		
5	Left semitendinosus (L.ST)	Medial 2/3 of the sciatic tuberosity to the upper tibia	80% from the sciatic tuberosity to the upper end of the tibia
6	Right semitendinosus (R.ST)		
7	Left gastrocnemius muscle (L.GM)	Internal popliteal fossa to internal Achilles tendon, highest point of tiptoe	Medial tendon to 90% of medial root bone
8	Right gastrocnemius muscle (R.GM)		

**Table 3 ijerph-20-01363-t003:** Shooting scores for participants wearing shooting shoes and sports shoes (MEAN ± SD) (units/scores).

Subjects	Shooting Shoes	Sports Shoes	*p*
S1	8.6 ± 1.7	9.7 ± 0.8	0.065
S2	9.0 ± 1.3	9.1 ± 0.9	0.835
S3	10.0 ± 0.5	9.1 ± 0.7	0.042 *
S4	9.9 ± 0.3	9.1 ± 0.8	0.060
S5	9.0 ± 1.0	8.9 ± 0.9	0.891
S6	9.2 ± 1.0	9.1 ± 1.1	0.835
S7	9.7 ± 0.9	8.0 ± 1.2	0.018 *
S8	9.6 ± 0.6	9.8 ± 0.6	0.225
Average	9.37 ± 1.09	9.10 ± 1.04	0.011 *

Note: Differences in comparison are indicated by * *p* < 0.05

**Table 4 ijerph-20-01363-t004:** Trajectory of center of pressure (COP) movement during use of shooting shoes and sports shoes (Mean ± SD).

Subjects	X (mm)		Y (mm)	
	Shooting Shoes	Sports Shoes	Shooting Shoes	Sports Shoes
S1	0.38 ± 0.10	0.67 ± 0.36	0.18 ± 0.31	0.43 ± 0.41
S2	0.43 ± 0.16	0.33 ± 0.15	0.17 ± 0.05	0.45 ± 0.41
S3	0.53 ± 0.22	0.67 ± 0.53	1.10 ± 1.57	0.20 ± 0.03
S4	0.48 ± 0.16	0.82 ± 0.24	0.72 ± 0.77	0.24 ± 0.11
S5	0.58 ± 0.35	0.62 ± 0.19	0.22 ± 0.06	0.29 ± 0.10
S6	0.47 ± 0.13	0.55 ± 0.15	0.23 ± 0.10	0.97 ± 1.37
S7	0.61 ± 0.38	0.49 ± 0.10	0.25 ± 0.09	0.34 ± 0.22
S8	0.50 ± 0.20	0.43 ± 0.38	0.17 ± 0.06	0.26 ± 0.07
Average	0.50 ± 0.21	0.57 ± 0.29	0.38 ± 0.61	0.40 ± 0.51
*p*-value	*P_x_* = 0.362		*P_y_* = 0.911	

**Table 5 ijerph-20-01363-t005:** A summary of trends in lower extremity muscle median frequency (MDF) during shooting and sports shoe use (Mean ± SD).

Title 1		TA	MRF		ST	
		Left	Left	Right	Left	Right
Shooting shoes	D1	66.2 ± 8.6	101.2 ± 26.7	83.7 ± 10.9	-	-
	D30	68.4 ± 8.7	105.1 ± 27.7	86.8 ± 16.9	-	-
	D60	67.7 ± 9.3	99.8 ± 25.7	81.7 ± 15.5	-	-
Sports shoes	D1	88.6 ± 35.2	125.9 ± 28.0	108.7 ± 34.0	88.0 ± 18.5	75.0 ± 22.6
	D30	84.4 ± 20.0	125.2 ± 27.5	111.7 ± 42.7	90.8 ± 23.2	73.1 ± 15.2
	D60	83.2 ± 12.0	119.2 ± 23.4	106.6 ± 36.6	90.1 ± 17.1	72.9 ± 14.6

Note: D1 indicates first shots; D30 indicates 30th shots; D60 indicates 60th shots; TA indicates tibialis anterior muscle; MRF indicates musculus rectus femoris; and ST indicates semitendinosus.

## Data Availability

The data presented in this study are available on request from the corresponding author.

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
