# Peer review of "A Comparative Study of the Fatigue of the Lower Extremities According to the Type of Shoes Worn When Firing a 10 m Air Pistol"

_ijerph, 2023, doi:10.3390/ijerph20021363_

Round 1

Reviewer 1 Report

I congratulate authors for their study dedicated the fatigue of the lower extremities according to type of shoes in air pistol shooters. Despite the excellent presentation of the results obtained there are some comments on this. Reading the manuscript I feel confused. Firstly, all abbreviations must be deciphered. Secondly, please, make the concordance between text, figures and tables. To be more specific, figure and numbers did not match numbers in the text. Lastly, the last paragraph in discussion section ‘Authors should discuss the results and how they can be interpreted from the perspective of previous studies and of the working hypotheses. The findings and their implications should be discussed in the broadest context possible. Future research directions may also be highlighted.’ is unnecessary

Author Response

Dear reviewer,
Thank you for your review. Please check the revised version manuscript.

Reviewer 2 Report

The introduction is riddled with grammatical errors, punctuation errors, repeated words, etc.

Would repeating the experiments with the same subjects on different days influence the results in any manner? Did the authors make any such attempts?

Figures/diagrams are not publication quality. Need to be substantially revised. Axis labels are missing in plots. The numbers on the axes are illegible.

It is advisable to add a limitations section to explain the constrains of the design of experiment.  

Author Response

(The authors gave the same response as above.)

Reviewer 3 Report

The topic of the paper is very interesting and current, both from the point of view of the author's profession and from the point of view of ergonomics. However, before the final release, I suggest some minor changes:

The introductory part is interesting; however, I suggest that you expand the introductory part with studies on this topic, from an ergonomic aspect.

Are all subjects healthy and have normal anthropometric measurements?

How did you choose the t-test? Have you checked the normality of the data distribution?

The results are very nicely presented and interesting. What worries me about the results is the reliability, based on a small sample. Add this fact to the limitation of the study.

The discussion is very transparent. However, there is a lack of studies to compare your results with.

Expand the conclusion with the following sections: directions for future research, limitations of the study, and practical application of the results.

Please expand the literature list with more recent studies.

Author Response

(The authors gave the same response as above.)

Round 2

Reviewer 2 Report

Minor typographical errors, must be rectified prior to the publication 

Reviewer 3 Report

Dear Authors,

The paper has significantly improved.